# A Multi-Hop LoRa Linear Sensor Network for the Monitoring of Underground Environments: The Case of the Medieval Aqueducts in Siena, Italy

**DOI:** 10.3390/s19020402

**Published:** 2019-01-19

**Authors:** Andrea Abrardo, Alessandro Pozzebon

**Affiliations:** Department of Information Engineering and Mathematical Sciences, University of Siena, 53100 Siena SI, Italy; alessandro.pozzebon@unisi.it

**Keywords:** M2M, D2D, LoRa, multi-hop, linear sensor networks, underground monitoring

## Abstract

In this paper, a pervasive monitoring system to be deployed in underground environments is presented. The system has been specifically designed for the so-called “Bottini”, i.e., the medieval aqueducts dug beneath the Centre of Siena, Italy. The results of a measurement campaign carried out in the deployment scenario show that the transmission range of LoRa (Long Range) technology is limited to a maximum of 200 m, thus, making the adoption of a classical star topology impossible. Hence, a Linear Sensor Network topology is proposed based on multi-hop LoRa chain-type communications. In this scenario, an ad-hoc transmission scheme is presented that optimally evaluates the wake-up time of all nodes with the aim of minimizing the average energy dissipation deriving from clock offsets. Numerical results show that the proposed wake-up time optimization leads in the best case to a 50% reduction in power dissipation with respect to a scheme that evaluates the wake-up time in a non-optimal way.

## 1. Introduction

Underground environments are one of the most challenging scenarios for the deployment of pervasive monitoring infrastructures. Apart from the intrinsic difficulties arising from the positioning of items in remote places, the two most critical factors to be addressed are energy supply and data communication. To this regards, monitoring solutions for some environments, like mines or underground railways, provided with power supply and wired connections for data transfer may be easily set up. However, in many cases this kind of infrastructure is not available. This is the case of the scenario discussed in this paper, i.e., the medieval aqueducts of the city of Siena, Italy, called the “Bottini”, that will be shortly described in Section 2.

As a matter of fact, in the absence of electrical power supply, powering a device in an underground environment is considerably more complex in comparison with an open-air site. Indeed, common energy-harvesting solutions like solar cells and wind turbines are useless due to the lack of the primary source of energy (i.e., the sun light and/or the wind). Hence, some recent works have investigated the possibility of deploying alternative energy-harvesting solutions [1,2]. However, off-the-shelf devices that incorporate these mechanisms are still difficult to be found.

At the same time, the problem of data transmission is even more challenging. Indeed, since a wireless connection with the surface is almost technically unfeasible, a wired connection with the underground environment, or a gateway, localized at one of the openings (e.g., the manholes) is mandatory to provide connection with the outside world. Then, provided that a wired connection (Internet, serial…) is brought to the underground, either a wired or a wireless network architecture is required to receive data from the large amount of devices that must be deployed to enable pervasive monitoring.

Wired-based communications would require the deployment of a huge amount of wirings, due to the large dimensions of the environment to be monitored. Moreover, the critical environmental conditions in terms of humidity, presence of water and the subsequent risk for flooding, would require a high quality, and then expensive, wired infrastructure. Finally, the deployment of such a kind of infrastructure in a site of high cultural value, like the one under study, would lead to an unacceptable negative aesthetic impact. For all these reasons, wireless communication is the only viable solution for the scenario at hand.

With that clear understanding, a problem that remains is the possibility of employing a long-range radio technique based on a standard star topology, where the nodes connect directly to the gateway without intermediate hops. Indeed, the propagation conditions of underground environments structured in galleries are typically characterized by strong power losses that considerably limit the radio coverage.

### Proposed Contribution and Related Works

In the context of Wireless Sensor Networks (WSNs) and monitoring infrastructures for underground environments, most of the existing works have focused on mine [3] and pipeline [4,5] scenarios. All the systems proposed in these papers focus on the realization of reliable wireless data transmission infrastructures in the presence of severe wireless propagation conditions that ask for the implementation of multi-hop linear networks. The concept of LSN has then emerged [6,7]. In this scenarios, the sensing devices are arranged according to linear or semi-linear layout where each node is part of a communication chain that requires each node to transmit its own data and/or forward the packets transmitted by the adjacent nodes.

According to [6], LSNs can be classified following either a Topological or a Hierarchical point of view. From the Topological point of view, they can be classified according to three typologies: *thin*, *thick* or *very-thick*. *Thin* LSNs are composed by a single chain arranged in one-dimensional fashion. *Thick* networks envisage the presence of a linear backbone that is able to collect data from Basic Sensor Nodes (BSN) randomly placed in the surroundings of the so-called Data Relay (DRN) and Data Dissemination (DDN) Nodes. Finally, in *Very Thick* Networks several parallel linear paths exist. In this case, every node can transmit in all directions to the surrounding nodes while the information flows linearly. The Hierarchical classification foresees the division of LSN in three categories: *One-Level*, *Two-Level* and *Three-Level*. in *One-Level* networks only BSNs are present. *Two-Level* networks allow the presence of both BNSs and DDNs while DRNs are present only in *Three-Level* networks.

Due to their importance for different application scenarios, wireless sensor networks based on LSNs have been considered in several studies. As an example, Chen and Wang [8] proposed a complete architecture for Chain-Type WSNs based on a hierarchical structure, discussing possible protocols both at a PHY and MAC level. An improvement of this architecture embedding the concept of Clustering is discussed in [9]. Other studies focus on single technical aspects of LSN. Yoo and Kim [10] propose an analytical method to evaluate the maximum throughput in general-purpose chain-type wireless networks, while a methodology to calculate the connectivity probability in thin one-dimensional LSNs is provided in [11]. The case of ZigBee-based LSN implementation is discussed in Sarr et al. [12], where a methodology for nodes’ auto-discovery and a strategy for ad-hoc hierarchical addressing is proposed. Furthermore, [13] focuses on the problem of scheduling while [14] proposes a routing algorithm customized to this scenario. While all the previous papers present interesting solutions to optimize parameters regarding data transmission in LSNs (throughput, scheduling, routing, etc. …), none of them deals with the problem of energy efficiency and then with the adoption of duty-cycling policies like the ones described in this paper.

Nevertheless, power consumption is an important line of research that has been discussed in several papers [15,16,17,18,19,20,21]. In particular, a MAC-level protocol customized to the LSN scenario is proposed in [15], while a TDMA-based scheduling algorithm for LSNs is discussed in [16]. Similarly, Toma et al. [17] propose a HDLC protocol for LSNs based on a TDMA MAC scheme, relying on the IEEE 802.15.4 PHY layer, and compare its performances with the ones of similar, widely adopted protocols. Zimmerling et al. [18,19] propose an energy efficient routing protocol for LSN. More specifically, a Minimum Energy Relay Routing (MERR) and an alternative adaptive version of the same protocol (AMERR) are proposed, respectively. Finally, some other papers focus on identifying the ideal topology and placement scheme for sensor nodes to maximize energy efficiency [20], also adopting transmission range adjustment policies [21]. All these papers present interesting solutions to increase energy efficiency in WSNs, however, few of them focus on the use of duty-cycling policies [15]. Anyway, in these works, no solution is adopted to adapt the activation and de-activation times of the nodes according to the synchronization errors and delays due to clock offsets and inaccuracies. We strongly believe that, in networks with very low sampling rates, the adoption of efficient duty-cycling policies and then the prolonged sleeping of the nodes, is the best way to reduce at minimum energy consumption.

Hence, we consider a multi-hop thin Linear Sensor Network (LSN) architecture based on the LoRa technology for pervasive monitoring of the underground “Bottini” environment, where data delivery with very low sampling rate can be tolerated. In this scenario, the main goal is that of prolonging the network lifetime as much as possible. While some works discussing the application of multi-hop LoRa can be found in the literature [22,23], to the best of our knowledge no solution has been proposed in the specific context of LSNs. More specifically, we leverage on the peculiarities of the considered scenario, to establish a duty cycling policy with long sleeping periods, alternated by relatively short transmitting periods. Several duty-cycle MAC protocols can be found in the literature, which can be roughly categorized into two categories: synchronous and asynchronous. In synchronous approaches, such as S-MAC [24], T-MAC [25], RMAC [26], and DW-MAC [27], neighbor nodes establish a common time scheduling. This goal is obtained by dividing the time into three periods, namely, *Synch*, *Data* and *Sleep*. In particular, during the Synch period, nodes exchange the respective scheduling times with the goal of waking up at the same time in the subsequent Data periods, thus, minimizing the energy wasting due to idle listening and overhearing.

In this paper we adopt the idea of dividing the time into Synch, Data and Sleep periods. In this setting, we propose a message exchange mechanism tailored to the LoRa based LSN scenario at hand, allowing each node to establish a common scheduling with adjacent nodes. In particular, the establishment of common scheduling is facilitated by the linear topology of the network, in which each node is simply requested to adopt the scheduling of the previous node in the chain. Moreover, we address the problem of possible node desynchronization after the sleeping periods. As a matter of fact, a fundamental assumption supporting synchronous approaches is that a separate protocol is used to synchronize the clocks during the Sync periods with required precision, e.g., by properly correcting the clock offsets, so that after a sleeping period adjacent nodes start the Synch operations at the same time. Several synchronization protocols could be considered to this aim that assume the presence of a deterministic trend characterizing the synchronization offsets among nodes (e.g., see [28,29]). However, internal clocks of commercial micro-controllers present random clock offsets that are in many cases impossible to predict, and that can be only statistically characterized. In these situations, maintaining time synchronization after long sleeping periods remains a big challenge. To circumvent such difficulties, we propose an asynchronous mechanism for exchanging the scheduling times during Synch periods, where tx–rx synchronization is achieved employing low power listening (LPL) techniques [30]. In this setting, assuming a given statistics of the clock error, we propose an analytical framework based on density evolution that optimally evaluates the wake-up time of all nodes after a Sleep period with the aim of minimizing the average energy dissipation deriving from clock offsets.

The rest of the paper is organized as follows. Section 2 presents the deployment scenario, i.e., the “Bottini”, focusing on the problem of data transmission in this kind of environment. Section 3 describes the configuration of the LoRa Network for this scenario, while in Section 4 the proposed multi-hop protocol is presented and an analytical framework for deriving the optimized wake-up times of nodes is proposed. Conclusive remarks are then drawn in Section 5.

## 2. The “Bottini” of Siena

The so-called “Bottini” are the medieval aqueducts that still feed the most part of the historic fountains in the Centre of the City of Siena, Tuscany, Italy. They are constituted by a series of galleries dug below the centre of the city and its close neighborhoods (see Figure 1a). The whole network is around 25 km long, and is composed of two main branches named “Bottino Maestro di Fonte Gaia” and “Bottino Maestro di Fontebranda” (from the names of the two monumental fountains respectively fed by the two branches) and a large number of secondary channels of different lengths. The structure was built from the XIII to the XV century and it is still perfectly preserved in its central sections while it is affected by different levels of degradation in the peripheral portions.

The dimensions of the galleries are roughly the same for the most part of the network: their cross section is characterized by a 170 cm average height and a 90 cm average width. For the most part the galleries are directly dug in the tuff stone that is present below the whole city, while some sections have been covered with bricks to prevent possible collapses. The water flows in a small open-air duct dug in the floor of the galleries: its cross section is roughly 20x8 cm and its slope is about 0.1%. While the path of the “Bottini” mainly runs from North to South, the galleries are not perfectly straight. indeed, they are characterized by a large number of slight curves that make the line-of-sight unachievable along the whole path.

Along the way, several inscriptions (like the one shown in Figure 1b) can be found on the wall that giveprecious historical evidence about the use of the aqueduct by the noble families of the city. Nevertheless, they are at a great risk of disappearing due to the harsh environmental conditions. Indeed, the “Bottini” system presents very high levels of humidity, while periodic flooding occurs due to heavy rains, thus, threatening the stability of the whole structure. For this reason, a pervasive real-time monitoring infrastructure is of great importance for the maintenance of the whole structure. Due to the peculiarity of the environment, the number of parameters to be monitored can be potentially very high. These concern both air and water and include the following:For air:–Temperature and humidity;–Currents and flows;–Concentration of gases (with a specific focus on radon which is radioactive and present in underground environments);For water:–Temperature;–pH;–Electrical conductivity;–Oxidation-Reduction Potential, (ORP);–Dissolved oxygen;–Turbidity;–Flow;–Presence of dissolved gases;–Concentration of Ca, Na, Mg, K, chlorides, sulfates, carbonates and organic compounds.

All these parameters affect the preservation of the environment and the safety of people and, hence, should be kept under control. In particular, they should be measured in different spots along the way and collected in real time in order to allow promptly intervention in case of critical situations. Nevertheless, a high sampling rate is not necessary because all the parameters of interest are not expected to present fast variations. Ideally, a hourly sampling rate suffices the requirements.

In the “Bottini” the electricity grid is not present nor can it be deployed. Sensing nodes should be powered by batteries and, accordingly, an important requirement is the power consumption in order to avoid frequent battery replacements. Similarly, since no wired connection is possible, a wireless data transmission infrastructure must be considered. Moreover, the connection with the surface is possible only at a limited number of spots (at the entrances): this means that, in order to cover all the structure, a wireless local area network is mandatory.

To sum up, according to the guidelines given by geologists and architects in charge of its preservation, a monitoring infrastructure for this environments should satisfy the following requirements:Low data sampling and transmission rate (hourly or even lower);Battery powering with consumption optimization in order to ensure long life to the sensor nodes;Wireless data transmission to one or more gateways positioned at the entrances;Coverage area in the order of some kilometers, with nodes placed at least every 200 m. Due to the presence of several wells and side galleries, the environmental conditions can change within few hundreds of meters, especially in terms of temperature and air currents thus, requiring the deployment of a pervasive monitoring infrastructure.

### Data Transmission Performances in the “Bottini” Environment

Basing on the above discussion, a data transmission technology suitable for the realization of the “Bottini” monitoring system must satisfy three main requirements: it has to be low power, it must enable multi-hop communications, and it must guarantee a minimum transmission range of 200 m. Accordingly, the most suitable solution can be identified among the Sub-GHz technologies that offer superior propagation properties compared to higher frequencies, e.g., the 2.4 GHz ISM bands [31]. In this context, LoRa technology has drawn a large interest and represents one of the most widely employed solutions, thus, enabling the use of a wide range of off-the-shelf devices.

LoRa technology is a Semtech’s proprietary technology [32] that is specifically aimed at the realization of low power monitoring systems. A similar proprietary solution currently available on the market is the Sigfox [33] technology. However, while SigFox technology exploits a global proprietary infrastructure and requires a subscription to transmit data to a cloud infrastructure, LoRa allows to set up private networks without any kind of subscription. For this reason, LoRa is provided with the adequate degree of flexibility to fit the needs of the specific application at hand.

More specifically, in order to minimize the power consumption of each nodes, LoRa networks are typically arranged according to a star topology. In this setting, a sensor node can wake-up, sample the data from the sensors, transmit them and go back to the sleep state, regardless of the behavior of the other sensor nodes. Then, upon receiving data from each node, the gateway (that is typically power supplied from the grid) collects and forwards them to the remote data management center.

In order to identify the best system configuration for the “Bottini”, the performances of LoRa were tested on the field, considering the transmission range as the main performance indicator since power consumption is only marginally affected by the environmental conditions.

The LoRa test infrastructure was composed of the following devices:An Arduino UNO board provided with a Libelium SX1272 LoRa module, connected through an ad-hoc Multiprotocol Radio Shield, powered by 4 1.5 V AA lithium batteries, in charge of sampling data from a test sensor (in this case an LM335 temperature sensor by Texas Instruments);A Waspmote USB Gateway provided with an SX1272 module and connected to a laptop to check in real time the actual data transmission.

The settings for the LoRa modules, chosen in order to achieve the largest transmission range, are shown in Table 1.

An important characteristic of the considered LoRa radio interface is the possibility of achieving very low sensitivity levels of the receiver (−137 dBm), when compared with possible alternatives. As an example, the different radio modules at 868 MHz present sensitivities in the −100 dBm/−120 dBm range [34]. On the other hand, lower sensitivity entails higher transmission range for a given transmission power, that is a fundamental requirement in the considered scenario.

The data transmission range was first tested outdoor in an urban environment. In this case, a transmission range of around 3 km was experienced: this value is in line with the ideal performances expected for the system. In a second step, we carried out experiments in the “Bottini” scenario. For this purpose, a section of around 500 m length of the so-called “Bottino Maestro di Fonte Gaia”, i.e., the gallery bringing the water to the fountain in the main square of the city (“Piazza del Campo”), was chosen. Figure 2a shows the network of galleries below the city centre while Figure 2b highlights the chosen section. It is worth noting that, while the gallery runs along a single direction (from south to north), the path is not perfectly straight. Several small curves prevent from having line-of-sight for more than 20–30 m.

The transmission range was measured in different points of the gallery, placing the devices right in the middle of the gallery floor. Figure 3a shows one of the test devices (the LoRa sensor node) positioned on the gallery floor while Figure 3b shows the transmission range measurement operations. As a result of these tests, the maximum achieved transmission range was around 200 m. Hence, a LoRa infrastructure with nodes placed at a distance in the range 100–200 m complies with the system requirements, prior of the implementation of a multi-hop protocol. Basing on this consideration, the LoRa technology was definitely adopted as a possible solution for the proposed scenario.

## 3. Nodes’ Configuration and Transmission Protocol

Despite LoRa allows to considerably increase the radio coverage with respect to alternative solutions, the standard star topology cannot be applied to the scenario at hand. Indeed, the overall length of the “Bottino Maestro di Fonte Gaia” is around 15 km, whereas the transmission range of the LoRa modules in the “Bottini” environment is around 200 m, thus, forcing the choice towards a Linear Network with multi-hop communications.

In particular, in the proposed architecture each sensor node is deployed at a distance of nearly 150 m each other. Hence, due to the limitations in the transmission range, each sensor node can directly communicate with adjacent nodes only. In this setting, a node acts simultaneously as an End Device transmitting its own data and as a Router forwarding the packets sent by the previous nodes, forming a chain from the first node down to the Gateway.

To implement this kind of transmission protocol, the firmware of each node must be able to receive and transmit. LoRa allows to send packets either to a specific node through its own address, or in broadcast modality using address 0. Since in the proposed configuration each node is directly connected only to 2 nodes (the previous and the following ones), a natural choice leans towards the choice of broadcast modality. Indeed, the broadcast packet can be received by the subsequent node, but also by the previous one. This opportunity can be exploited to set up an overhearing mechanism which will be described in the following Section, with low power acknowledgement modality.

It is worth noting that the proposed network topology is not intrinsically fault-tolerant. Indeed, a malfunctioning in one node blocks the transmission from all the previous nodes. Nevertheless, this chain-type data transmission makes the identification of the malfunctioning node very simple. Indeed, in the case a node is damaged, the Gateway will receive packets starting from the next node in the chain only. Hence, since the positioning of each node is known a priori, the malfunctioning node will be then easily identified. Due to the use of the broadcast modality, the substitution of the malfunctioning node will be totally transparent: the new node must simply be deployed in place of the old one without any network re-configuration.

In the proposed network architecture, only two typologies of nodes are then possible: the Gateway node (Figure 4a) and the Sensor Node (Figure 4b). The Gateway node is the terminal node of the Linear Network and it is in general unique: it is only in charge of receiving all the packets and forwarding them to the data collection centre. However, it is the most complex node due to the presence of two types of connectivity. Hence, the Gateway to be deployed in the “Bottini” will be composed of a microcontroller, a LoRa module and a GSM module for data transmission.

The Sensor Nodes have basically the same setting of the device used during the tests (see Section 2.1). The most significant difference is the presence of a larger number of sensors: the first prototype includes an air and humidity sensor and a water temperature sensor. As for the transmission modality, all nodes adopt the same parameters presented in Table 1, including the SF fixed at 12 in order to achieve the maximum possible range. Note that, since SF is set to 12 for each node, it is not necessary to communicate this parameter to adjacent nodes. The nodes are able to receive and transmit and, hence, to enable Peer-to-Peer communications. More specifically, the proposed multi-hop procedure is implemented by creating an ad-hoc firmware deployed on the microcontroller of each node.

In order to optimize the overall system power consumption, a duty cycling policy has been adopted. To elaborate, the Sensor Node is *active* only for the amount of time required for sensing the environment and for data communication. This time will depend on several factors, such as the number of packets to be transmitted (considering both the locally generated and the forwarded traffic), the transition time from *sleep* to *active* mode and viceversa, and the time needed to acquire synchronization among nodes. On the other hand, following the Semtech SX1272 LoRa module datasheet [32], the transition times account for a maximum of few ms, and hence it can be neglected in the following.

As for transmission time, the Time on Air for a LoRa module with the settings defined in Table 1, i.e., CR = 4/5, SF = 12 and BW = 125 kHz, can be calculated according to [35], and it has been evaluated in different conditions in [36]. For a 51 byte payload it has the following value:(1)$$T_{onAir}≃2.1\;s$$

Concerning power consumption, the SX1272 datasheet provides the following values:Supply current in Receive Mode (Boost modality off)
(2)$$I_{RX}=10.5\;\mathrm{mA}$$Supply current in Transmit Mode (Boost modality off, RFOP =+13dBm)
(3)$$I_{TX}=28\;\mathrm{mA}$$

Nevertheless, these values do not take into account the power consumption of additional components like the microcontroller, the sensors and the connection circuitry. Accordingly, in order the get a more realistic value, the actual current absorption has been experimentally acquired by powering the test Sensor Node with a laboratory DC power supplier and then measuring the current with a digital multimeter. The following values have been measured:Supply current in Receive Mode (Boost modality off)
(4)$$I_{RX}=66\;\mathrm{mA}$$Supply current in Transmit Mode (Boost modality off, RFOP =+13 dBm)
(5)$$I_{TX}=98\;\mathrm{mA}$$

These values show that the overall consumption of the logic and acquisition part of the Node would be unsustainable if the nodes were always active. As an example, for a commercial battery capacity of 3–4 Ah, the battery should be recharged every 1–2 days. On the other hand, in order to implement an efficient duty cycling mechanism, the clocking mechanism is of particular importance. To this respect, two possible clocking systems can be considered for the scenario at hand: (*i*) an internal clock (natively embedded inside a microcontroller) or (*ii*) an external clock (generally an ad-hoc RTC or any other kind of device as for example a counter [37]). While external clocks are in general more accurate [38], internal clocks prevent from the necessity of using an additional device. Moreover, with internal solutions, the length of the sleep period can also be dynamically changed through the use of an ad-hoc firmware. For this reason, the use of the internal clock has been adopted in this work.

Needless to say, one of the most important factors that affect the efficiency of duty-cycling mechanisms in the considered scenario is given by the clock accuracy. This aspect will be extensively discussed in the next Section. In order to estimate the accuracy of the internal clock of a microcontroller, a test duty-cycling routine was set up on an Arduino UNO board. In this routine a 3600 s sleep period was set: after this lapse of time, the microcontroller wakes up and transmits a packet. The Timestamp of the packets arrival are recorded and the differences between two successive Timestamps are taken as a measure of the actual sleeping period. A total of 1000 values was measured and their distribution is shown in Figure 5, where $f_{\delta }\left(x\right)$ is the pdf of the sleeping time error. It is worth noting that, despite the clock error is unbiased, a considerable deviation with respect to the expected waking up time is observed (in certain cases the error can be higher than 100 s).

In order to put in evidence the critical role of the synchronization inaccuracies in the considered scenario, a prototype monitoring system has been set up and tested in the laboratory. In particular, we have implemented an architecture composed of 2 Sensor Nodes and one Gateway Node arranged in a chain fashion: such a configuration allows to set up a a multi-hop connection between one Sensor Node (from here on called the End Node-EN) and the Gateway (GW), exploiting the second Sensor Node as the intermediate hop (from here on called the Intermediate Node-IN). The hardware architecture of the nodes is the same as the one adopted for the LoRa transmission range tests, for both the Sensor Nodes and the GW. Regarding the software, the IN node has been provided with an ad-hoc firmware aimed at realizing the routing functionality that is not available in the standard LoRa protocol. In particular, we have implemented the receiving and transmitting functionalities in the same IN node, whereas standard LoRa nodes are configured to exclusively transmit or receive packets.

To further elaborate, both nodes are kept active for a limited span of time after which they go back into sleeping mode for a fixed amount of time. Conversely, the GW is kept always active since in the final configuration it is expected to be powered by the electricity grid. Duty-cycling is implemented exploiting the internal clock of the microcontrollers. More specifically, the EN node is activated every $T_{EN}$ seconds to transmit a single packet. As for the IN node, it is activated every $T_{IN}$ seconds and, upon activating, it goes into listen mode for a fixed period: if it receives a packet, it forwards it to the GW and then it goes back into sleeping mode. In the case of ideal internal clocks, the sleeping period $T_{IN}$ of the IN could be calculated so that the IN node wakes up at the exact time the EN node transmits its packet, i.e., since the IN node goes to sleep $T_{onAir}$ seconds later than the EN node, we have $T_{IN}=T_{EN}-T_{onAir}$. However, owing to synchronization errors, the IN node should wake-up in advance in order to prevent situations in which the EN node transmits while the IN node is still sleeping. Accordingly, we have set $T_{IN}=T_{EN}-4$ s, i.e., with a time advance of nearly 2 s with respect to the ideal case.

We have then collected results for three different $T_{EN}$ values, i.e., $T_{EN}=120$ s, $T_{EN}=300$ s and $T_{EN}=600$ s. In all situations, we have transmitted 100 packets and we have measured the number of packets received by the GW. The results are shown in Table 2.

It is worth noting that in the considered setting, $T_{EN}=120$ s is the maximum allowable sleeping period to prevent synchronization loss between the nodes. With such a relatively low sleeping period the IN node would consume nearly 4 mAh every hour, with 30–40 days of battery duration, that is definitely unsustainable in the scenario at hand. Of course, in the presence of a longer network with more than one packet to be delivered towards the GW, the situation will only get worse. In order to increase the sleeping periods up to 1 h, that is acceptable for the Bottini monitoring system, it is necessary to devise an ad-hoc synchronization and data transmission protocol, that is the subject of the next Sections.

## 4. The Considered Nodes’ Synchronization and Data Transmission Protocol

The proposed network architecture is composed of a set of *S* sensor nodes arranged as a chain, with sensor node 1 transmitting to the adjacent node 2 which will then forward the packet to node 3 an then on and on in a similar fashion down to node S+1 which acts as a gateway forwarding the received packets to a data management centre. While the topology is fixed, the position of each node in the network is known a priori: this means that also the role of each node is known in advance.

### 4.1. Synchronization Protocol

The presence of considerable clock offsets (see Figure 5) imposes the adoption of a synchronization protocol to make data transmission possible. As matter of fact, the clock offset leads to a misalignment in the waking up instants of two adjoining nodes: this means that one node may wake-up while the following is still sleeping, and then the transmitted packet is inevitably lost. Hence, we propose a synchronization protocol where, as in classical synchronous MAC protocols, the time is divided into SYNCH, DATA, and SLEEP periods. The SYNCH period is used to communicate the DATA transmission scheduling to the next node in the chain, following a SYNCH packets flooding mechanism, where a SYNCH packet is transmitted from node 1 to node *S* in the chain.

Before discussing the proposed synchronization protocol, we first notice that in this setting a mechanism for informing the source node that the SYNCH packet is correctly received is mandatory. To this aim, LoRaWAN protocol provides an ACK mechanism that, however, requires the transmitter to remain active for the time of the reception of a whole additional packet [39]. In order to further optimize the power consumption, we propose instead an acknowledge mechanism based on a Low Power Listening (LPL) approach.

To elaborate, we first note that, owing to the particular environment considered in this paper, each node can retransmit the SYNCH packet as soon as it receives it without the risk of incurring in a collision. Hence, the transmitter knows that, upon correctly receiving the SYNCH packet, the subsequent node immediately transmits it towards the next node in the chain. Accordingly, the transmitter may remain active for a short period *overhearing* the channel to understand whether the next node transmits or not.

The proposed procedure can be summarized as follows: node s−1 broadcasts its SYNCH packet and steps down in LPL modality, overhearing the channel soon after it terminates the transmission. Node *s* receives the SYNCH packet and then forwards it to node s+1 in broadcast modality; such transmission is detected also by node s−1. Nevertheless, node s−1 does not receive the whole packet: instead, after receiving the first byte it immediately starts the sleeping period, thus, reducing power consumption with respect to the classical ACK mechanism. In case node s−1 did not detect the transmission of the SYNCH packet, it would retransmit it continuously, as in classical LPL schemes.

Assume now that at a given initial time all nodes are synchronous. Then, after a supposedly long sleeping period, denoted by $D_{long}$, *s*-th node in the chain should ideally schedule its re-activation with a given delay $R_{s}$ with respect to node 1, to give the previous nodes time to transmit all the SYNCH packets. Ideally, node *s* would wake-up at the exact moment that node s−1 starts its transmission. However, as a consequence of inevitable clock offsets, the exact delay time $r_{s}$ when each node actually activates after the sleeping period will be different from $R_{s}$. In particular, denoting by $\delta_{s}$ the synchronization error for node *s*, we have $r_{s}=R_{s}+\delta_{s}$, where the entity of $\delta_{s}$ will be strictly related to $D_{long}$. Hence, a protocol for managing clock misalignments is required, so that packets can be successfully delivered to the Gateway and, at the same time, nodes are prevented from long idle listening that would rapidly determine node’s batteries depletion.

In the proposed approach, the SYNCH procedure is initiated by node 1. In particular, node 1 sets $R_{s}=0$ and, upon activating, transmits a SYNCH packet towards the second node using the overhearing and LPL functionalities described before. The SYNCH packet contains three control fields, namely, the elapsed time (ET), the number of packet to be forwarded (FP), and the *short* reactivation time to forward the data packets ($D_{short}$).

ET is a counter reporting the number of SYNCH packets transmitted in the chain and it is initialized to zero by node 1. Denoting by $T_{p}$ the fixed duration of a packet (including overhearing), ET can be expressed in terms of number of slots of duration $T_{p}$. Then, upon receiving the SYNCH packet from node 1, node 2 increases the value of ET by one, and starts transmitting the SYNCH packet towards node 3. ET is then increased by one at every unsuccessful packet transmission. This mechanism is repeated until the SYNCH packet is received by node *S*. We denote by ET(s) the ET value seen by node *s*. Note that the knowledge of ET allows each node to know the exact time the SYNCH operation has been terminated by node 1, and hence to re-synchronize its wake-up schedule with node 1. FP is a counter indicating the number of packets that each node will have to forward in the DATA period. Hence, in the transmission chain from node 1 to node *S*, FP is increased by one by each node having a packet to transmit. Accordingly, the FP value in the SYNCH packet received by node *s*, say it FP(s), contains the number of packets it will have to forward during the data transmission operation.

As for the scheduled activation times $R_{s}$, in the absence of synchronization errors the natural choice would be to set

$R_{s}=0$, for s=2, and $R_{s}=\left(s-2\right)\left(T_{p}-T_{OH}\right)$, for s>2, $T_{OH}$ being the overhearing period. Indeed, in this case node 2 activates at time 0 and can receive the SYNCH packet from node 1, node 3 activates at time $T_{p}-T_{OH}$ and receives the SYNCH packet from node 2, and so on down to node *S*. Instead, in the presence of synchronization errors this is not the optimal choice, as it will be shown in the sequel.

Figure 6 sketches the flow of the SYNCH packets for the first 6 nodes following node 1, underlining the increase in the FP and ET values with the progress of the synchronization. In the Figure the wake-up times $r_{s}$ and the correspondent different nodes’ modality, namely, idle listening, SYNCH and data packets reception (RX), SYNCH packet transmission (SYNCH TX) and overhearing are reported.

### 4.2. Data Transmission Protocol

The data transmission phase is initialized by the first node in the chain that has a packet to transmit, say it node $s_{0}$. To elaborate, after a short sleeping period denoted by $D_{short}$ following the successful transmission of the SYNCH packet, node $s_{0}$ wakes up and transmits the DATA packet. In general, $D_{short}\left(s\right)$ represents the time that elapses between the end of SYNCH packet delivery and the start of data packet transmission at node *s*.

To elaborate, $D_{short}\left(s_{0}\right)$ is set equal to a predetermined time interval $\Delta_{S}$ that is long enough to guarantee that the next node has terminated the synchronization procedure with high probability and, accordingly, it is ready to receive a packet. Farther this paper gives an insight on how to determine this term in order to guarantee a minimum success probability. In order to avoid abuse of notations, we still denote by $R_{s}$ the scheduled re-activation time of each node after the transmission of the SYNCH packet, i.e., $R_{s_{0}}=\Delta_{S}$. Then, the procedure for delivering all the required data packets to the Gateway is explained below for the generic couples of nodes *s* and s+1, with $s\geq s_{0}$.

Upon receiving $D_{short}\left(s\right)$ from node *s*, node s+1 may calculate the waking up time for receiving data packets from node *s*, i.e.: (6)$${\tilde{R}}_{s+1}=D_{short}\left(s\right)-\left(ET(s+1)-ET(s)\right)$$

Note that ET(s+1)−ET(s) gives the delay between node s+1 and node *s* in the synchronization chain. Moreover, node s+1 is able to compute the reactivation time $D_{short}\left(s+1\right)$ for forwarding the FP(s+1) data packets. To this aim, node s+1 first evaluates $D_{short}^{\prime }\left(s+1\right)$ as:(7)$$D_{short}^{\prime }\left(s+1\right)={\tilde{R}}_{s_{1}}+FP\left(s\right)T_{p}^{\prime }$$that represents the time instant when node s+1 will have received all the FP(s) packets from node *s*, where $T_{p}^{\prime }=T_{p}-T_{OH}$ is the duration of a data packet. Hence, the actual $D_{short}\left(s+1\right)$ is computed as:(8)$$D_{short}\left(s+1\right)=\max\left(D_{short}^{\prime }\left(s+1\right),\Delta_{S}\right)$$i.e., the actual $D_{short}\left(s+1\right)$ must be evaluated in order to be sure that the next node will have terminated the synchronization process. Afterwards, $D_{short}\left(s+1\right)$ is communicated to node s+2 through the SYNCH packet, and node s+1 goes back to sleeping mode.

Finally, node s+1 wakes up at instant $R_{s+1}={\tilde{R}}_{s+1}-T_{A}$, where $T_{A}$ is a timing advance that is necessary to compensate for possible small clock offsets that can be accumulated during the short sleeping period. In particular, $T_{A}$ must be set at least equal to the maximum possible clock offset, so that node s+1 will be for sure ready to receive when node *s* wakes up and starts transmitting the data packets.

Figure 7 sketches the flow of the DATA packets for the first 6 nodes following node $s_{0}$, referring to the synchronization phase reported in the example of Figure 6. In the Figure the correspondent different nodes’ modality, namely, idle listening, data packet transmission (PCK TX) and overhearing are reported.

### 4.3. Optimization of the Activation Times in SYNCH Mode

In this Section we derive an analytical framework for calculating the optimal activation times $R_{s}$ for SYNCH operation with the aim of minimizing the sum of energies consumed at each node. To this aim we consider a reference time of a SYNCH packet duration for the calculation of the consumed energies. Since the nodes are expected to be battery powered with an almost constant Voltage value, we can equivalently consider energy consumption or battery capacity depletion as performance metric. Hence, we introduce the following parameters:
$C_{tx}$: Battery capacity depletion for one packet transmission (including overhearing)$C_{lst}$: Battery capacity depletion for listening to the channel for a period of one packet duration (including overhearing)

Let us make the reasonable assumption that the synchronization errors are independent from each other, and denote by $f_{\delta }\left(x\right)$ the pdf of the synchronization error for a generic node. Given the above we have $f_{r_{s}}\left(x|R_{s}\right)=f_{\delta }\left(x-R_{s}\right)$.

We now propose a density evolution approach to evaluate the optimum wake-up times $R_{s}$ assuming that both $f_{\delta }\left(x\right)$ and $p_{s}$ are known. To this aim, denote by $f_{t_{s}}\left(x\right)$ the pdf of $t_{s}$, $t_{s}$ being the time when node *s* starts transmitting. For the sake of notation simplicity, $t_{s}$, $r_{s}$ and $R_{s}$ are assumed in the following to be normalized with respect to the packet duration $T_{P}$. Note that, since the reference node 1 starts transmitting as soon as it becomes active, we have $f_{t_{1}}\left(x\right)=f_{\delta }\left(x\right)$.

Let now denote by $Q_{tx,s}$ and $Q_{lst,s}$ the battery capacity depletion wasted by node *s* for transmitting in SYNCH mode before node s+1 is active and for listening the channel before node s−1 starts transmitting its SYNCH packet, respectively. We have: (9)$$\begin{array}{c} Q_{tx,s}=\left\{\begin{array}{cc} \left\lceil r_{s+1}-t_{s}\right\rceil C_{tx} & \mathrm{if}\;r_{s+1}\geq t_{s}\\ 0 & \mathrm{otherwise} \end{array}\right.\\ Q_{lst,\begin{array}{c} s+1 \end{array}}=\left\{\begin{array}{cc} \left(t_{s}-r_{s+1}\right)C_{lst} & \mathrm{if}\;r_{s+1}\leq t_{s}\\ \begin{array}{c} \left[\left\lceil r_{s+1}-t_{s}\right\rceil -\left(r_{s+1}-t_{s}\right)\right]C_{lst} \end{array} & \mathrm{otherwise} \end{array}\right. \end{array}$$where ⌈•⌉ is the ceiling operator, i.e., $\left\lceil r_{s+1}-t_{s}\right\rceil$ is the number of transmitted packets and overhearing periods of node *s* before node s+1 is ready to receive. We are now in the position of evaluating the wasted energies in (Equation 9) as a function of $R_{s+1}$ as: (10)$$\begin{array}{c} Q_{tx,s}\left(R_{s+1}\right)=C_{tx}\int_{-\infty }^{\infty }\int_{-\infty }^{y}\begin{array}{c} \left\lceil y-x\right\rceil \end{array}f_{r_{s}}\left(y|R_{s}\right)f_{t_{s}}\left(x\right)dxdy\\ Q_{lst,s+1}\left(R_{s+1}\right)=C_{lst}\int_{-\infty }^{\infty }\int_{-\infty }^{x}\left(x-y\right)f_{r_{s+1}}\left(y|R_{s}\right)f_{t_{s}}\left(x\right)dxdy+\\ \begin{array}{c} C_{lst}\int_{-\infty }^{\infty }\int_{-\infty }^{y}\left[\left\lceil y-x\right\rceil -\left(y-x\right)\right]f_{r_{s}}\left(y|R_{s}\right)f_{t_{s}}\left(x\right)dxdy \end{array} \end{array}$$

Since the other sources of energy consumption, i.e., the energies due to transmission and reception of data packets, do not depend on $R_{s}$, the value of $R_{s+1}$ that minimizes the overall weighted consumed energies can be set as:(11)$$\begin{array}{c} R_{s+1}=\underset{\begin{array}{c} R \end{array}}{\arg\;\min\;}\left[w_{s}Q_{tx,s}\left(R\right)+w_{s+1}Q_{lst,s+1}\left(R\right)\right] \end{array}$$where the set of weights $w_{s}$ can be chosen to enforce a given degree of fairness among nodes. The solution of (Equation 11) can be computed through numerical computation of (Equation 10) for each possible value of $R_{s+1}$. It is worth noting that in order to evaluate all the $R_{s+1}$ for s=1,…,S−1 in an iterative fashion, it is necessary to propagate the knowledge of $f_{t_{1}}\left(x\right)$ to subsequent nodes.

To this aim, it is straightforward to derive: (12)$$\begin{array}{c} f_{t_{s+1}}\left(x\right)=f_{t_{s}}\left(x-1\right)\mathrm{Prob}\left\{r_{s+1}\leq x-1\right\}+\begin{array}{c} \left(\sum_{n=2}^{\infty }f_{t_{s}}\left(x-n\right)\right)\mathrm{Prob}\left\{x-2\leq r_{s+1}\leq x-1\right\} \end{array}\\ =f_{t_{s}}\left(x-1\right)\int_{-\infty }^{x-1}f_{r_{s+1}}\left(y|R_{s+1}\right)dy+\begin{array}{c} \sum_{n=2}^{\infty }f_{t_{s}}\left(x-n\right)\int_{x-2}^{x-1}f_{r_{s+1}}\left(y|R_{s+1}\right)dy \end{array} \end{array}$$where the two terms in (Equation 12) correspond to the situations where $r_{s+1}\leq t_{s}$ and $r_{s+1}>t_{s}$, respectively. Note that in the second case, *n* represents the number of SYNCH packets transmitted by node *s* before node s+1 may transmit its own packet.

#### Total Battery Capacity Depletion

We are now in the position of evaluating the total average battery capacity depletion for each transmission cycle due to SYNCH and DATA transmission operations. To this aim, let us assume that, at each transmission cycle, each node produces a data packet with a given transmission probability $p_{tx}$. Hence, we observe that according to the protocol depicted in Figure 7, the capacity depletion at node *s* for transmitting data after synchronization is due (on average) to three terms: (*i*) idle listening of $T_{A}$ seconds (only if s<S); (*ii*) reception of $\left(s-1\right)p_{tx}$ data packets from node s−1 (only if s>1); (*iii*) transmission of $sp_{tx}$ data packets towards node s+1. Accordingly, the total average capacity depletion can be derived as $Q_{tot,s}=Q_{synch,s}+Q_{data,s}$, where:
(13)$$\begin{array}{c} Q_{synch,s}=\left\{\begin{array}{cc} Q_{tx,s}T_{P}+C_{tx} & \mathrm{if}\;s=1\\ Q_{tx,s}T_{P}+C_{tx}+Q_{lst,s}T_{P}+C_{lst} & \mathrm{if}\;1<s<S\\ C_{tx}+Q_{lst,s}T_{P}+C_{lst} & \mathrm{if}\;s=S \end{array}\right.\\ Q_{data,s}=\left\{\begin{array}{cc} p_{tx}C_{tx} & \mathrm{if}\;s=1\\ sp_{tx}C_{tx}+\left(s-1\right)p_{tx}C_{lst}+T_{A}C_{lst} & \mathrm{if}\;1<s<S\\ sp_{tx}C_{tx}+\left(s-1\right)p_{tx}C_{lst} & \mathrm{if}\;s=S \end{array}\right. \end{array}$$

### 4.4. Inter-Node Synchronization Time Statistics

We are now in the position of deriving the statistics of the inter-node synchronization time $\delta_{s}$, which represents the time needed for completing the SYNCH phase at node s+1 starting from the time $T_{S}$ of correct SYNCH packet delivery at node *s*. Note that according to the above notations, we have $T_{S}=t_{s+1}+T_{O}$, and $\delta_{s}=t_{s+2}-T_{S}+T_{O}=t_{s+2}-t_{s+1}$. Hence, $\delta_{s}$ is an integer multiple of the packet duration $T_{P}$, with $P_{\delta_{s}}\left(n,t\right)=\mathrm{Prob}\left\{\delta_{s}=nT_{P}\left|t_{s+1}=t\right.\right\}$, with n≥1, given by: (14)$$\begin{array}{c} P_{\delta_{s}}\left(n,t\right)=\left\{\begin{array}{cc} \mathrm{Prob}\left\{r_{s+2}\leq t\right\}=\int_{-\infty }^{t}f_{r_{s+2}}\left(y|R_{s+2}\right)dy & \mathrm{if}\;n=1\\ \mathrm{Prob}\left\{\left(n-2\right)T_{P}<r_{s+2}-t\leq \left(n-1\right)T_{P}\right\}=\int_{t+\left(n-2\right)T_{P}}^{t+\left(n-1\right)T_{P}}f_{r_{s+2}}\left(y|R_{s+2}\right)dy & \mathrm{for}\;n\geq 2 \end{array}\right. \end{array}$$

Hence, the unconditioned distribution $P_{\delta_{s}}\left(n\right)$ can be evaluated as:(15)$$\begin{array}{c} P_{\delta_{s}}\left(n\right)=\int_{-\infty }^{\infty }P_{\delta_{s}}\left(n,t\right)f_{t_{s+1}}\left(t\right)dt \end{array}$$

To further elaborate, the minimum waiting time interval $\Delta_{S}=mT_{P}$ at node *s* to guarantee that node s+1 has terminated the SYNCH procedure can be evaluated by imposing a predetermined target success probability $P_{S}$, i.e., *m* is the minimum integer such as $\sum_{k=1}^{m}P_{\delta_{s}}\left(k\right)\geq P_{S}$. Of course, owing to the random nature of the synchronization errors, it is always possibile that node s+1 cannot terminate the SYNCH phase before node *s* starts transmitting its data packets. In this case, node s+1 completes the SYNCH procedure and soon after starts listening to the channel to receive the remaining data packets. Then, the missed packets can be replaced with dummy packets in order to keep the structure of the protocol intact.

### 4.5. Results and Comparisons

The performances of the wake-up time optimization scheme proposed in this Section are analyzed in terms of average consumed battery capacity and per-node consumed battery capacity for each SYNCH operation. More specifically, we consider the power consumption values reported in Section 3, i.e., a current drawn of 98 mA in transmission mode and 66 mA in receiving and listening mode. Hence, neglecting the energy spent for overhearing, we get for a given packet duration of 2.1 s:
(16)$$\begin{array}{cc} C_{tx}=\frac{98\times 2.1}{3600}\;\mathrm{mAh}=0.0572 & \;\mathrm{mAh} \end{array}$$
(17)$$\begin{array}{cc} C_{lst}=\frac{66\times 2.1}{3600}\;\mathrm{mAh}=0.0385 & \;\mathrm{mAh} \end{array}$$

As for the synchronization error statistics, we consider the clocking system reported in Section 3.

In this setting, we compare the performances in terms of consumed battery during SYNCH operations of the proposed optimized wake-up time procedure with a non-optimized one where the wake times are determined assuming perfect synchronization among nodes, i.e., $R_{s}=\left(s-2\right)T_{p}$. Finally, the set of weights $w_{s}$ of the optimized scheme are set to one.

The cases N=5 and N=10 are shown in Figure 8a,b, while the cases N=20 and N=50 are shown in Figure 9a,b. In all cases, we report the per-node and the average battery capacity depletion during SYNCH operations for the optimized and non-optimized schemes. In order to definitely assess the validity of the analysis, we have also carried out a simulator that implements the proposed SYNCH and DATA protocols depicted in Figure 7. The average battery capacity depletion during SYNCH operations obtained through simulations, reported as diamond markers in Figure 8a and Figure 9b, show a perfect matching with the analysis.

These results demonstrate the validity of the proposed wake-up optimization scheme that allows to clearly reduce the average consumed battery power per node with respect to the non-optimized scheme in all the considered scenarios. In particular, it is shown that the average reduction of battery capacity depletion is more evident with the increase of the number of nodes, going from a minimum of 10% for N=5 to around 50% for N=50.

As for the per-node battery capacity depletion during SYNCH operations, it is worth noting that in the non-optimized case it increases with the order of the nodes in the transmission chain. This is because for each pair of transmitting and receiving nodes the time when SYNCH transmission can be successfully carried out depends on the maximum between the two respective synchronization errors, since both nodes must be active at the same time. Accordingly, despite the clock error is unbiased, the maximum between two independent clock errors it is not and, hence, the wake-up time of the receiver tends to be more and more in advance with respect to the transmission time of the transmitter. This *lagging-behind* effect ultimately increases the idle listening periods of the nodes with the increase of their order. Of course, this effect is not present in the optimized scheme that allows to achieve a quasi-equalization charge of consumed battery power among all nodes, thus, allowing to noticeably reduce the consumed battery power of high order nodes. In particular, considering the case N=50, the most power hungry node (node S−1) in the optimized case saves around 65% of battery power with respect to the non-optimized one. It is worth noting that in all cases node *S* consumes less battery power than node S−1. This is because node S+1 is always active and, hence, node *S* transmits the SYNCH packet only once. Moreover, it is also worth noting that in the optimized scheme the low order nodes can consume more battery power than in the non-optimized scheme (e.g., this is the case of node 2 in all the considered scenarios). This is due to the fact that in order to minimize the average consumed battery power it might be convenient to increase the battery capacity depletion of a node, e.g., by increasing its average idle listening period.

In order to better highlight the *lagging-behind* effect in the non-optimized case, we report in Figure 10 the optimized and non-optimized $R_{s}$ for N=50. It can be observed that the optimal $R_{s}$ increases more than linearly, and the difference between the optimal and non-optimal $R_{s}$ increases with the node order. As further consideration, it is worth noting that the optimization scheme will inevitably increase the time needed for achieving synchronization, that is not however a critical parameter in the scenario at hand.

As final results, we report in Figure 11a,b the total battery capacity depletion for $p_{tx}=0.5$ and for $T_{A}=T_{P}$. This choice for the timing advance has been verified to conservatively compensate for possible small clock offsets accumulated during the short sleeping period. As for $\Delta_{S}$, this parameter is set following the procedure described in Section 4.4 by setting $P_{S}=0.995$ which yields, for the scenario at hands, $\Delta_{S}=30T_{P}$. The average battery capacity depletion $Q_{tot,s}=Q_{synch,s}+Q_{data,s}$ evaluated as in (Equation 13) is compared with actual total average capacity depletions derived through simulations for both the optimized and non optimized SYNCH procedures. The cases N=10 and N=20 are shown in Figure 11a,b, respectively, showing an almost perfect matching between analytical and simulation results. A similar behavior is observed for different parameters setting. The actual probability $P_{O}$ of missing a packet in the transmission chain due to TX/RX misalignment has been estimated through simulations as $P_{O}\leq 0.05$ in all cases, in line with the imposed maximum $P_{S}=0.995$.

## 5. Conclusions

In this paper a LoRa-based monitoring system specifically designed for a specific underground scenario (i.e., the “Bottini” of Siena) is presented. The proposed solution can be also applied in a wide range of possible scenarios, from mines to pipelines, where Linear topology is the only viable option. In particular, an ad-hoc transmission scheme has been presented that optimally evaluates the wake-up time of all nodes with the aim of minimizing the average energy dissipation deriving from clock offsets. Numerical results show that the proposed wake-up time optimization leads in the best case to a 50% reduction in power dissipation required to acquire synchronization with respect to a scheme that evaluates the wake-up time in a non-optimal way. As next step, a real implementation of the system will be carried out to definitely assess the validity of the proposed protocol.

## Figures and Tables

**Figure 1 sensors-19-00402-f001:**
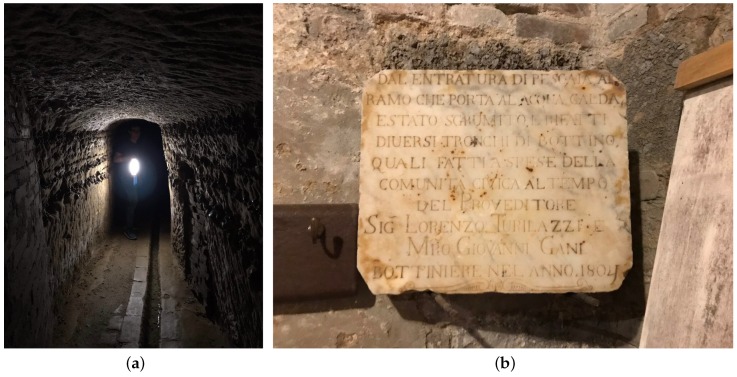
(**a**) A view of one of the “Bottini” galleries and (**b**) one of the historical inscriptions.

**Figure 2 sensors-19-00402-f002:**
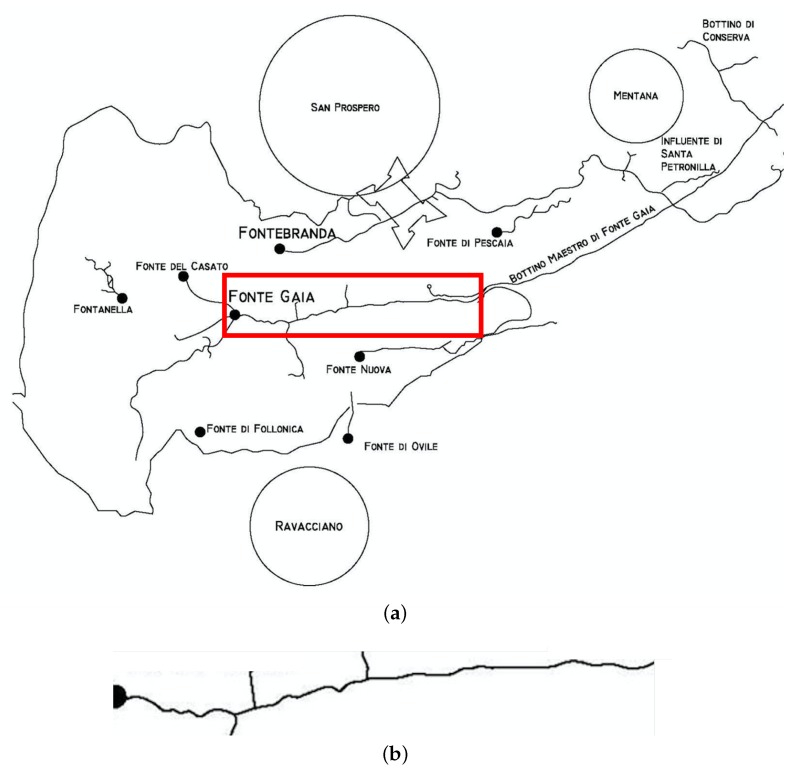
(**a**) Map of the “Bottini” galleries below the city centre of Siena, with the test section (marked in red) and (**b**) course of the gallery in the test section.

**Figure 3 sensors-19-00402-f003:**
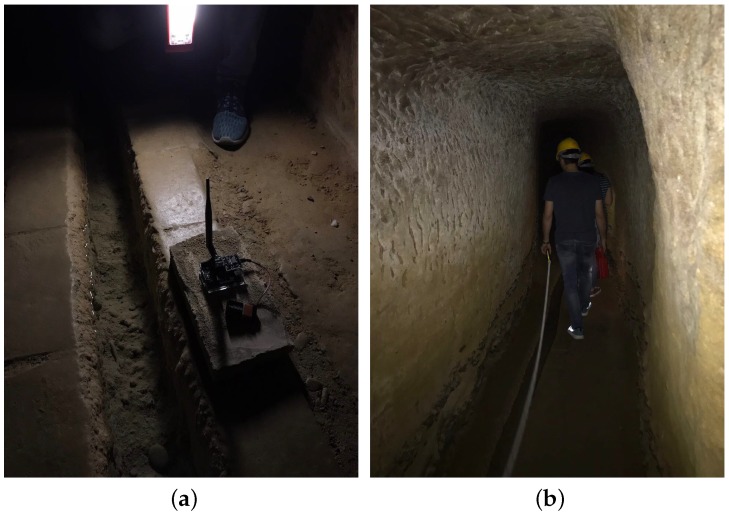
(**a**) The LoRa transmission module and (**b**) the data transmission range measurement operations.

**Figure 4 sensors-19-00402-f004:**
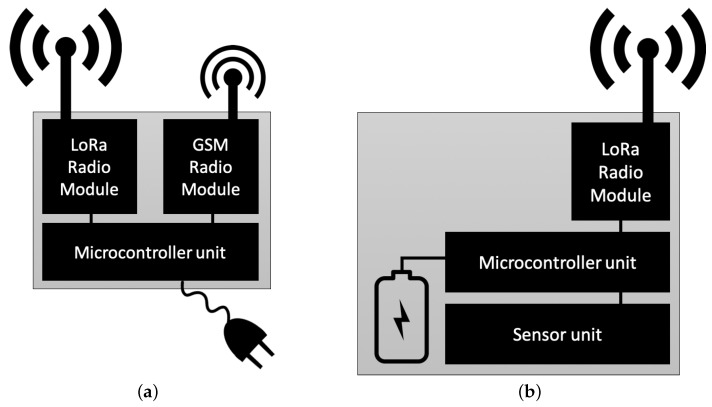
Architecture of (**a**) Gateway Node and (**b**) Sensor Node.

**Figure 5 sensors-19-00402-f005:**
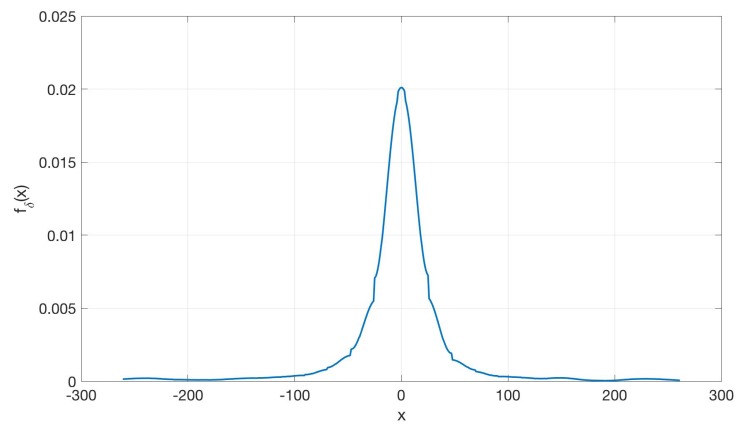
Distribution of the clock error.

**Figure 6 sensors-19-00402-f006:**
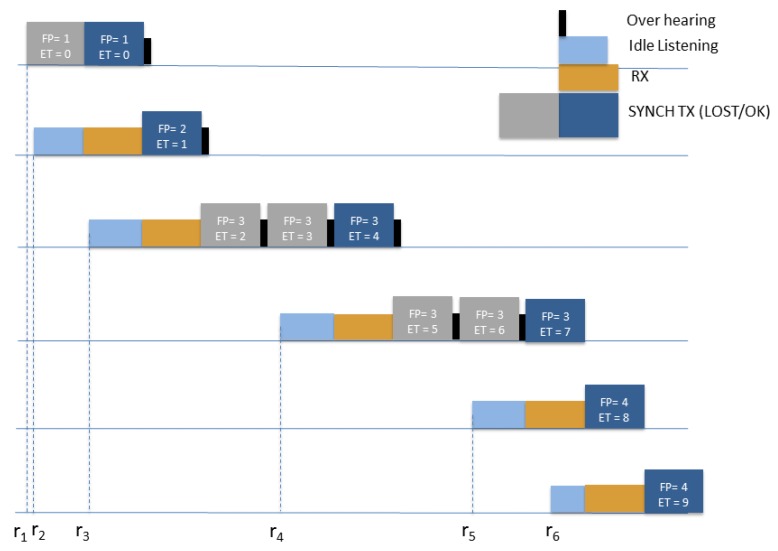
Illustrative example: synchronization protocol with the FP and ET values for the first 6 nodes.

**Figure 7 sensors-19-00402-f007:**
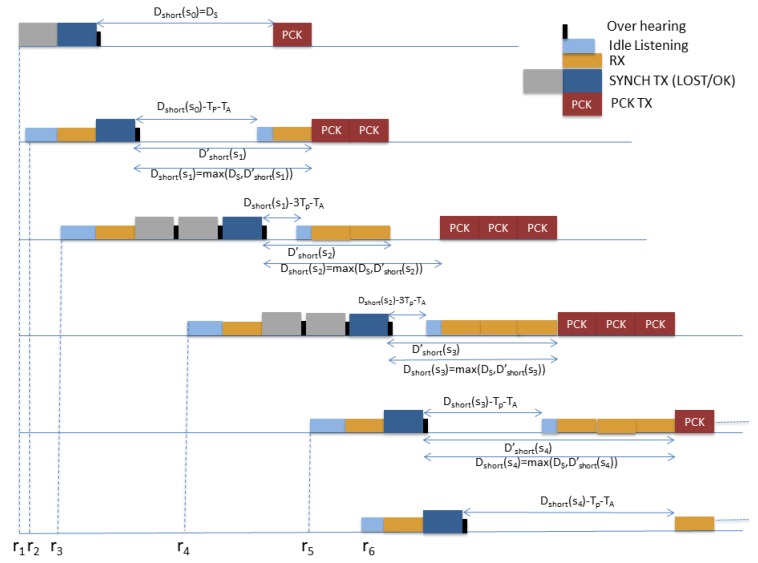
Illustrative example: flow of the DATA packets for the first 6 nodes (the synchronization phase is the same of Figure 6.

**Figure 8 sensors-19-00402-f008:**
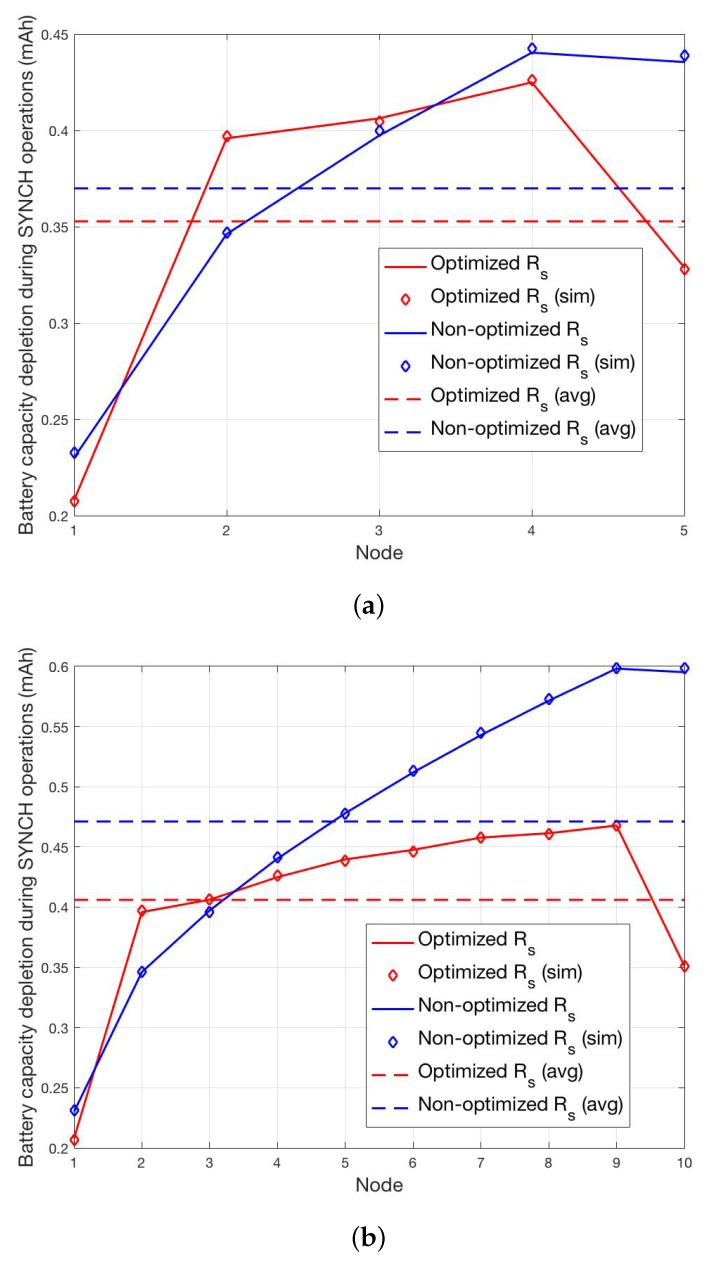
Per-node and average battery capacity depletion during SYNCH operations for the optimized and non-optimized schemes for N=5 (**a**) and N=10 (**b**).

**Figure 9 sensors-19-00402-f009:**
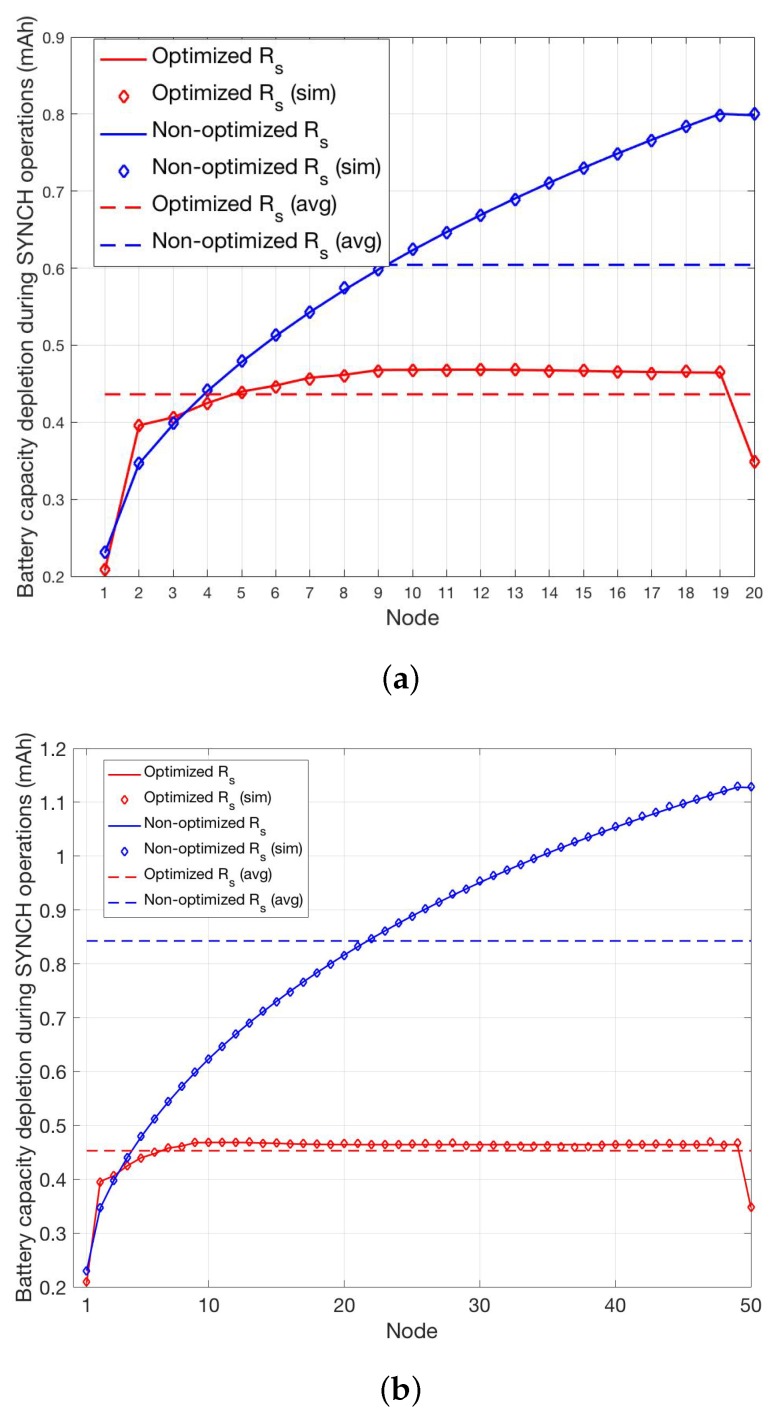
Per-node and average battery capacity depletion during SYNCH operations for the optimized and non-optimized schemes for N=20 (**a**) and N=50 (**b**).

**Figure 10 sensors-19-00402-f010:**
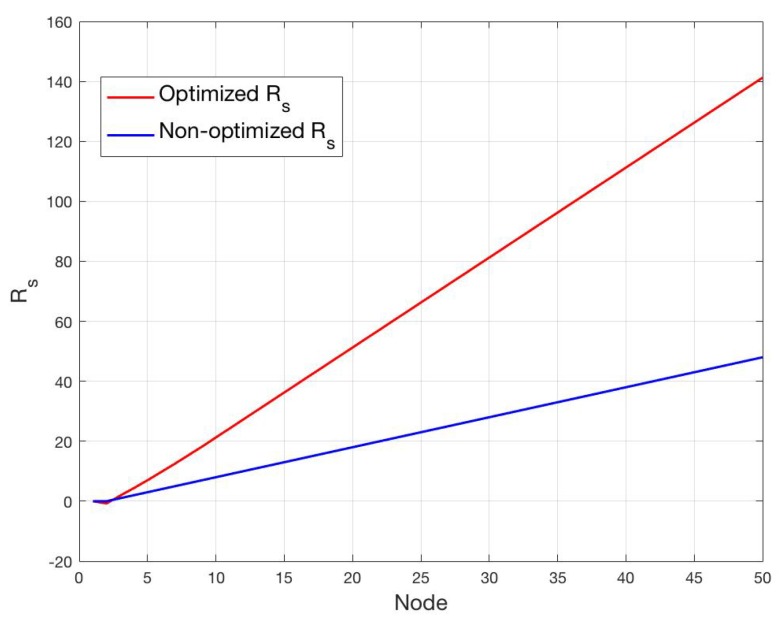
$R_{s}$ values for the optimized and non-optimized schemes for N=50.

**Figure 11 sensors-19-00402-f011:**
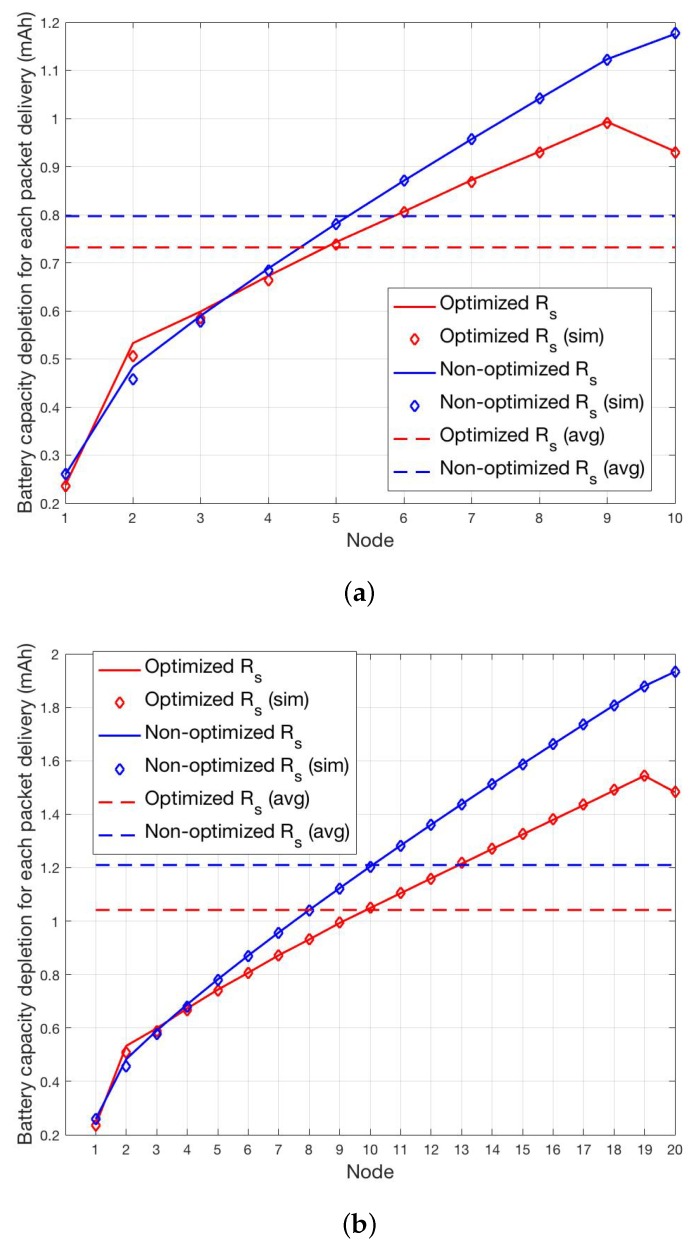
Per-node and average total battery capacity depletion for the optimized and non-optimized schemes for N=10 (**a**) and N=20 (**b**).

**Table 1 sensors-19-00402-t001:** LoRa module settings.

Parameter	Value
Center frequency	868.10 MHz
Bandwidth	125 KHz
Coding Rate	4/5
Spreading factor (SF)	12
Output power	14 dBm

**Table 2 sensors-19-00402-t002:** Multi-hop data transmission. Packet loss rate in different conditions.

EN Sleeping Period	IN Sleeping Period	Transmitted Packets	Received Packets	Packet Loss Rate
120 s	116 s	100	100	0%
300 s	296 s	100	72	28%
600 s	596 s	100	43	57%

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
