# Peer review of "A Multi-Hop LoRa Linear Sensor Network for the Monitoring of Underground Environments: The Case of the Medieval Aqueducts in Siena, Italy"

_sensors, 2019, doi:10.3390/s19020402_

Reviewer 1 Report

The problem described in the paper is very interesting. The authors present the part of the sensor network project for sensing underground environmental data with sensor nodes equipped with LoRa transceivers. The paper contains a very good introduction, description of the hardware and theoretical power consumption analysis. 

I have the following main remarks regarding the paper:

The authors propose their solution to the linear sensor network, the hardware used for the network is presented, as well as the synchronisation and data protocols. However, the work seems to be not finished, as the paper does not contain any experimental data from the operating network, only theoretical values and simulations are presented. The authors should add the results from the working sensor network regarding the real throughput and battery depletion.

The authors compare their protocol to the naive, artificial case of the basic protocol. In such a comparison, any addition of time synchronisation will result in the improvements. The authors should compare their solution to the existing sensor network protocols.

Minor remarks:

Page 7 line 223 (Section 3.)  The authors say that the first node only transmits, the other transmit and receive. However to use ACK, the receiving node has to also transmit.

The differences between the nodes have been described in the text, but for clarity, I would suggest to add the schematic diagrams of the various typos of the nodes, so the reader could immediately see the difference.

Page 10 lines 332-3368 the paragraph containing the description of the ET is not clear, I would suggest to rewrite it and add some examples or pictures to better explain the meaning of the ET.    

The paper should be checked agains spelling, I noticed some mistakes, e.g.:

page 2 line 63: “...and Three-Level. in One-Level networks....”

page 8 line 249: “... to communicate this parameters to ...”

page 9 line 280: “... of the packets arrival are recored and ...”

Reviewer 2 Report

 LoRa has not been used much. However, it has potential to solve many problems.

This paper presents a pervasive monitoring system to be deployed in underground environments. The system is designed for Bottini - the medieval aqueducts dug beneath the Centre of Siena, Italy. The results show that the transmission range of LoRa technology is limited to max of 200 m, thus making the adoption of a classical star topology impossible.

 Author Response

See attached file

Reviewer 3 Report

The authors proposed an ad-hoc synchronization protocol that optimally evaluate the wake up time of all nodes with the aim of minimizing the average energy dissipation from clock drifts in multi-hop LoRa Linear Sensor Network.

The comments are as follows:

1.Since there are some existing work on Linear Sensor Network, the authors should explain why those work is not suitable for this scenario.

2.Figure 5 shows the synchronization protocol with the FP and ET values for the first 5 nodes, however, this figure is not very clear.

3.In simulation, the proposed protocol should compare with other similar work instead of the naïve scheme that assumes perfect synchronization among nodes.

Author Response

See attached file

Round  2

Reviewer 1 Report

The authors significantly improved their paper, addressing most of the remarks. However, the main drawback of the paper is not reduced: the lack of the practical tests and measurement results of protocol operation. The authors present only simulation results. As the paper contains the detailed description of the hardware, it would seem obvious, that the implementation of the sensor network in real conditions should be presented. I suggest postponing the publication of the paper until the measurement results are ready, that would greatly improve the paper quality. I think that for a linear network a simple setup consisting of few nodes would be enough to prove the proposed solution.

The authors mentioned in their response, that there are no papers in the literature optimization of wake up times after long sleeping time is addressed. Here are some potential papers that I found mentioning this problem, which could be referenced:

 Ge Huang, Albert Y. Zomaya, Flávia C. Delicato, Paulo F.Pires, Long term and large scale time synchronization in wireless sensor networks, Computer Communications, Volume 37, 1 January 2014, Pages 77-91.

 Madhushri, Priyanka, Jovanov, Emil, Long-Term Synchronization of Hybrid Sensors Networks, International Journal of Embedded and Real-Time Communication Systems (IJERTCS), 2018.

 M. Elsharief, M. A. Abd El-Gawad and H. Kim, "FADS: Fast Scheduling and Accurate Drift Compensation for Time Synchronization of Wireless Sensor Networks," in IEEE Access, vol. 6, pp. 65507-65520, 2018.

 Wanlu Sun, E. G. Ström, F. Brännström and D. Sen, "Long-Term Clock Synchronization in wireless sensor networks with arbitrary delay distributions," IEEE Global Communications Conference (GLOBECOM), Anaheim, CA, 2012, pp. 359-364.

Qasim M. Chaudhari, Erchin Serpedin, "Clock Estimation for Long-Term Synchronization in Wireless Sensor Networks with Exponential Delays", Hindawi Publishing Corporation EURASIP Journal on Advances in Signal Processing, Volume 2008, Article ID 219458.

Author Response

See attached pdf

Round  3

Reviewer 1 Report

I still maintain my opinion that the article lacks measurement results from a functioning sensor network. I suggest giving the authors a few months to add experiment results.

Author Response

See attached pdf
